# Plug & play origami modules with all-purpose deformation modes

Chao Zhang[1,2,7], Zhuang Zhang[1,3,7], Yun Peng[4], Yanlin Zhang[1], Siqi An[1,3], Yunjie Wang[1], Zirui Zhai [5], Yan Xu [2] ✉ & Hanqing Jiang [1,3,6] ✉

Three basic deformation modes of an object (bending, twisting, and contraction/extension) along with their various combinations and delicate controls lead to diverse locomotion. As a result, seeking mechanisms to achieve simple to complex deformation modes in a controllable manner is a focal point in related engineering fields. Here, a pneumatic-driven, origami-based deformation unit that offers all-purpose deformation modes, namely, three decoupled basic motion types and four combinations of these three basic types, with seven distinct motion modes in total through one origami module, was created and precisely controlled through various pressurization schemes. These all-purpose origami-based modules can be readily assembled as needed, even during operation, which enables plug-and-play characteristics. These origami modules with all-purpose deformation modes offer unprecedented opportunities for soft robots in performing complex tasks, which were successfully demonstrated in this work.

Three basic motion types, namely, bending, twisting, and contraction/extension, as well as their combinations, constitute the primary locomotion of creatures in nature[1,2]. Thus, the tireless pursuit of mimicking these simple to complex motion types is the focal point in the robotics field. Among many attempts at these endeavors, soft actuators with inherent compliance and safer human–robot interaction abilities have drawn increasing attention, although existing designs typically exhibit unimodal motions, such as contraction[3-8], bending[8-11], and twisting[8,11-15], which hinders them from achieving motions with multiple degrees of freedom that are essential for complex tasks. Superposition of these unimodal motions via series connection[16-18] provides a straightforward means for complex spatial motions, albeit the inevitable complicity in both single motion trajectory and controls led by the simple function of each individual actuator. The development of versatile soft actuators, enabling richer motion types and more flexible motion trajectories, is therefore desired. Consequently, actuators with multimodal modes have emerged for coupled motion types (e.g., coupled twisting and

contraction)[19-28]. Among these structures of actuators, such as twisting tower[13,19], elastic sheets with kirigami patterns[20], local constraints[11,21-23], parallel spaced multiple chambers[24-26] and origami-inspired designs[21,27-31], such as Yoshimura[32,33], and Kresling origami[34,35] and Miura[36] represent a distinctive structural design method offering unique merits in actuation, including a high deformation ratio[36-38], multi-stability[30,39], and manipulative stiffness[40,41], although existing origami-based actuators still suffer limited multi-degree-of-freedom motions resulting from their original origami patterns. Decoupled all-purpose deformation modes have rarely been demonstrated in a single actuation module, seriously weakening the potential for dexterous manipulation and locomotion.

In this work, by breaking the energy-favorable deformation mode of the Kresling origami pattern (i.e., coupled twisting and contraction) through individually accessible, pneumatically driven pouches on the side of the pattern, a bending mode was added, and consequently, a two-level Kresling origami pattern with the opposite chirality was created as an all-purpose module. This module can realize three

[1]School of Engineering, Westlake University, Hangzhou, Zhejiang 310030, China. [2]Key Laboratory of Soft Machines and Smart Devices of Zhejiang Province, Zhejiang University, Hangzhou, Zhejiang 310027, China. [3]Westlake Institute for Advanced Study, Hangzhou, Zhejiang 310024, China. [4]School of Mechanical Engineering, Dongguan University of Technology, Dongguan, Guangdong 523808, China. [5]School for Engineering of Matter, Transport and Energy, Arizona State University, Tempe, AZ 85287, USA. [6]Research Center for Industries of the Future and School of Engineering, Westlake University, Hangzhou 310030, China. [7]These authors contributed equally: Chao Zhang, Zhuang Zhang. ✉e-mail: xyzs@zju.edu.cn; hanqing.jiang@westlake.edu.cn

decoupled basic motion types (i.e., bending, twisting, and contraction/extension) and four combinations of these three basic types, with seven distinct motion modes in total through one module (Fig. 1a). The present methodology overperforms other means (e.g., origami-based, simple pneumatic driven, and smart materials based soft actuators) in terms of the motion modes (Fig. 1b). Most actuators, including conventional electric motors or smart materials (Fig.1b), generally exhibit single motion mode. Even though some designs can achieve multiple modes through a single module, the available modes are still limited and generally coupled (Fig. 1b). In contrast, our design achieves the integration of all seven motion modes through barely one module. Ready assembly of these all-purpose Kresling origami-based modules, even during operation, empowers plug-and-play characteristics of these origami patterns, which greatly enables the capability of pneumatically driven robotics. These origami modules with all-purpose deformation modes offer unprecedented opportunities for soft robots to perform complex tasks.

## Results

### The design concept of the all-purpose origami module: a new bending mode

The Kresling origami pattern has drawn increasing attention in the robotics field due to its large contraction ratio and simultaneous twisting motion capability. Nevertheless, the existing designs are still strongly constrained by the geometric feature of the Kresling pattern, resulting in coupled twisting-contraction/extension under pneumatic actuation[42] and global deformation-based bending under the actuation of tendons[43] or a magnetic field[28]. To enrich the motion versatility of the original Kresling pattern, two new creases, namely, $AE$ and $CE$, with $E$ being the midpoint of $AC$ (Fig. 2a), were added, and a pouch was attached to the four-vertex unit $ABCD$ (marked in yellow) to form an inflatable cell with two states, i.e., a deflated state and an inflated state. The deflated state corresponds to the original Kresling pattern, while the inflated state introduces a new constraint, which breaks the energy-favorable path and introduces a new deformation mode. Aside from the pouch having deflated and inflated states, a main chamber formed by the original Kresling pattern was also constructed (see Methods), consisting of an airtight chamber and flexible creases. Thus, a pneumatic-driven Kresling pattern with a main chamber and side pouches was formed. The geometric design of the one-level origami structure can be seen in Methods, Supplementary Fig. 1, and Supplementary Fig. 2. The detailed fabrication processes and the

pneumatic actuation methods are provided in Supplementary Fig. 3, Supplementary Fig. 4, and Supplementary Fig. 5.

By inflating and deflating the side pouches, the Kresling pattern exhibits a coupled twisting and contraction motion (same as the original Kresling pattern) and a newly added bending mode. To describe the deformation modes, as controlled by the pressurization scheme, a binary BTC (bending-twisting-contraction) code is introduced with "1" for the activated mode and "0" for the deactivated mode, e.g., $\frac{BTC}{100}$ for a bending mode and $\frac{BTC}{101}$ for a coupled bending and contraction mode. To represent the pressurization scheme, $[P_m|P_p]$ is introduced, which represents the pressurization scheme for the main chamber ($P_m$) given that for the side pouches ($P_p$). A binary system, with "−1" for deflating, "0" for inactive (i.e., ambient pressure), and "1" for inflating, is adopted to describe how the pneumatic process is applied, and the detailed pressure is given in the text. The combined BTC code (i.e., XYZ) and the pressurization scheme define the deformation mode and the corresponding pneumatic actuation, i.e., $\frac{BTC}{XYZ}[P_m|P_p]$.

When the side pouch is in the inactive state (Fig. 2b), upon negative pressure $P_m$ applied in the main chamber, all the facets fold inward (Supplementary Movie 1). The experiments agree well with the theoretical analysis (see Methods and Supplementary Fig. 6) and finite element simulations (Supplementary Fig. 7). Thus, the inactive side pouch does not affect the original motion mode of a Kresling pattern, i.e., coupled contraction and twisting. When inflating the side pouch, the relatively high stiffness of the side pouch interferes with the original Kresling pattern, and a bending deformation appears upon deflating the main chamber (Fig. 2c). All the triangular facets, except those attached to the inflated side pouch, fold inward, leading to an inhomogeneous deformation (Supplementary Movie 2), namely, coupled bending and twisting deformation $\frac{BTC}{011}[-1|1]$, and the comparison between experiments and simulation is given in Supplementary Fig. 8. Due to the large bending angle (e.g., approximately 17.4°/21.7 mm in height for $P_m = -6$ kPa in the main chamber), the arc length (i.e., contour height) on the tensile side is used to characterize the height (see Supplementary Note 1 and Supplementary Fig. 9), which exhibits a 2.4% change. Notably, because of the constraint imposed by the inflated side pouch, the twisting angle $\beta$ is significantly smaller than that for the inactive side pouch (e.g., approximately 8.2° in Fig. 2c vs. 24.2° in Fig. 2b at the same $P_m = -2$ kPa). Given that the present pattern has three modes, namely, extension/contraction, twisting, and bending, although they are still coupled, this pneumatic-driven Kresling pattern forms the

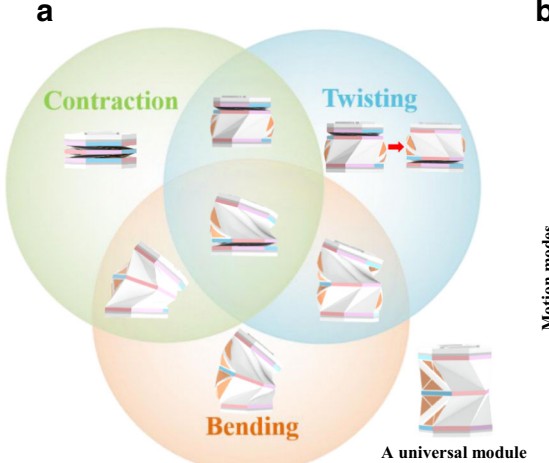

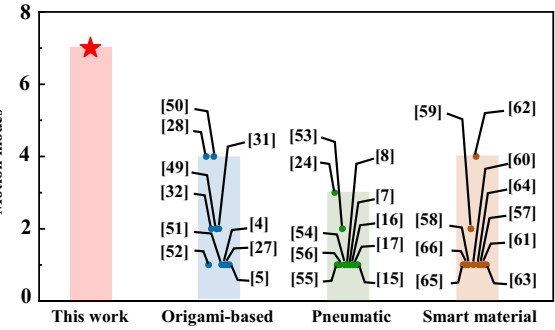

**Fig. 1 | All-purpose deformation modes of the present origami module. a** A universal actuation module consists of an appropriate combination of the Kresling origami patterns and inflatable cells. This module exhibits contraction/extension, bending, twisting, and any combination of these motion modes via pneumatic actuation. **b** Comparison of the motion modes of the present module with existing origami-based[4,5,27,28,31,32,49–52], pneumatic[7,8,15–17,24,53–56], and smart material actuators[57–66].

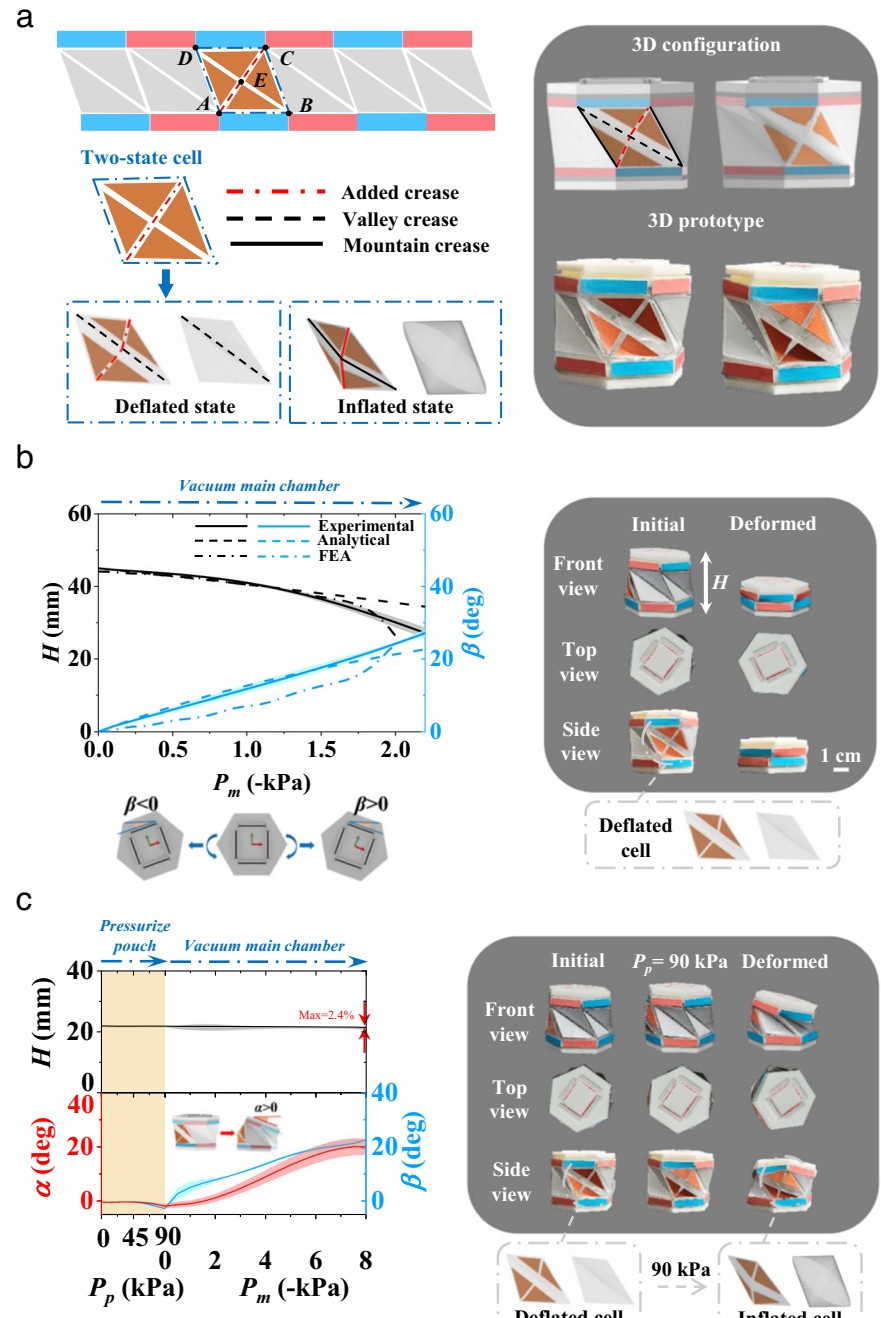

**Fig. 2 | Design concept and motion capability of a pneumatic-driven Kresling pattern with a side pouch. a** Integration of a side pouch into an airtight Kresling pattern. **b** Coupled twisting and contraction motion when the side pouch is deflated and the main chamber is vacuumed. The twisting angle $\beta > 0$ when the upper cap twists clockwise and $\beta < 0$ otherwise. **c** Coupled bending and twisting motion when a side pouch is inflated and the main chamber is vacuumed. The bending angle $\alpha > 0$ when the upper cap bends anticlockwise. Shaded regions represent 1 standard deviation; solid lines represent the mean values.

basis of the all-purpose origami module with various deformation modes, as discussed in the following.

## An all-purpose origami module with all deformation modes: construction and controls

A two-level Kresling pattern with side pouches and opposite chirality is constructed to form an all-purpose origami module capable of all deformation modes (Fig. 3a). The side pouches enable a bending mode when they are inflated asymmetrically, and the main chamber is deflated, and the two-level structure with the opposite chirality enables cancellable twisting deformation. In the following, we demonstrate its capability to achieve seven distinct deformation

modes, as controlled by the inflation/deflation scheme of the side pouches and the main chamber. A similar binary coding system is adopted here with a slight difference considering the side pouches on the top and bottom levels, $[P_m | {}^{P_p^t \leftrightarrow P_p^t}_{P_p^b \leftrightarrow P_p^b}]$, in which $P_m$, $P_p^t$, and $P_p^b$ present the pressurization scheme for the main chamber given that the side pouches on the top and bottom levels and $\leftrightarrow$ denotes the two side pouches on the opposite sides.

The pure contraction mode is achieved by keeping all the side pouches inactivated and deflating the main chamber, i.e., ${}^{BTC}_{001}[-1| {}^{0 \leftrightarrow 0}_{0 \leftrightarrow 0}]$, as shown in Fig. 3b and Supplementary Movie 3. During the entire contraction process, the height change agrees reasonably with the

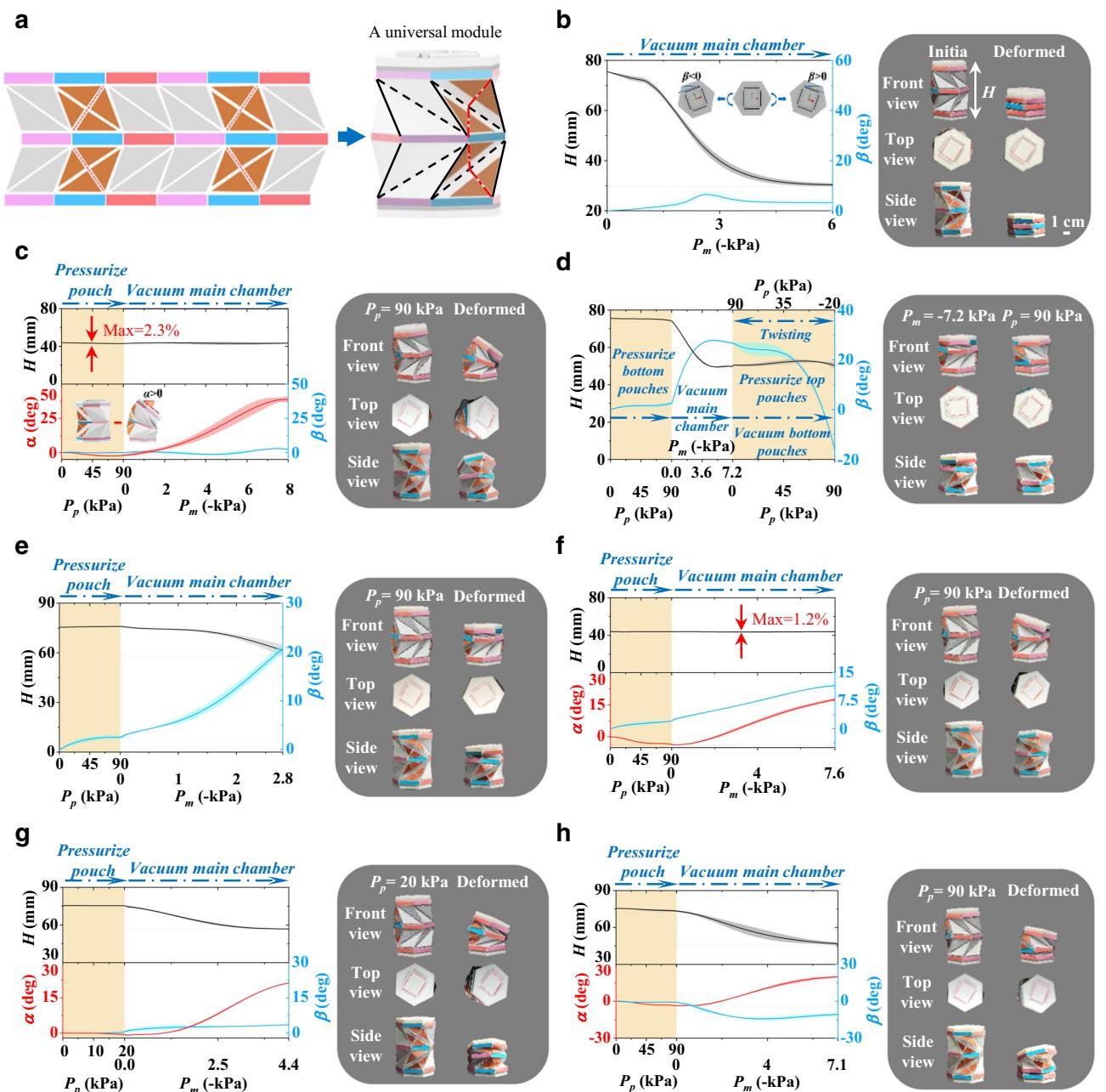

**Fig. 3 | Design concept and motion capability of an all-purpose origami module. a** Construction of a two-level Kresling pattern with side pouches and opposite chirality. The dashed-dotted lines, dashed lines, and solid lines represent added creases, valley creases, and mountain creases, respectively. **b–h** Seven distinct motion modes of the origami module, including contraction motion when no side pouch is inflated (**b**); bending motion when the two pouches on one side are inflated (**c**); twisting motion when the inflation/deflation of the pouches on the two levels is switched (**d**); coupled twisting and contraction motion when the pouches on the top level are deflated, while those on the bottom level are inflated (**e**); coupled bending and twisting motion when one pouch on the top level is deflated, while other pouches are inflated (**f**); coupled bending and contraction motion when the two pouches on one side are inflated with a lower pressure compared with bending motion (**g**); and all-three coupled motion when one pouch on the top level is inflated and the other pouches are deflated (**h**). Here, the main chamber is vacuumed for the actuation of all motion modes. Shaded regions represent 1 standard deviation; solid lines represent the mean values.

finite element results (Supplementary Fig. 10). The opposite chirality structure canceled out most of the twisting deformation, and the stabilized twisting angle of approximately 2.9° was due to the fabrication error, although the peak twisting angle was approximately 6.5° because of the imperfect synchronization of the opposite twisting motions on the top and bottom levels, which can be further optimized by precise manufacturing.

The bending mode is realized by inflating the side pouches on one side of the cylindrical structure to a certain pressure (e.g., 90 kPa in Fig. 3c and Supplementary Movie 4), followed by deflating the main chamber, i.e., $\frac{BTC}{100}[-1|\frac{1\leftrightarrow0}{1\leftrightarrow0}]$. The asymmetric inflation of the side pouches (e.g., $1 \leftrightarrow 0$) on both levels leads to bending when the main chamber is vacuumed, where the twisting deformation (see Fig. 3c) is canceled out from the opposite twisting direction. During the entire bending motion, the contour height and the twisting angle remain fairly unchanged (with approximately 2.3% and 2.98° variations, respectively), and the maximum bending angle can be as large as approximately 38.8°, which agrees reasonably well with the simulation results (Supplementary Fig. 11). The discrepancy in the bending angle is mainly due to the initial stress in the origami module during the

assembly process versus a perfect state in the finite element analysis. Notably, by inflating the side pouches on different sides, this module can bend in opposite directions.

Different from other deformation modes using the two-level unit as the reference, the initial state for pure twisting (Fig. 3d and Supplementary Movie 5) is a one-level unit achieved by this pressurization scheme $[-1|\frac{0\leftrightarrow0}{1\leftrightarrow1}]$, in which the bottom level is nondeformable due to pressurized side pouches up to 90 kPa and the top level is fully compressed by vacuuming the main chamber. Then, the main chamber is kept at its deflated state while simultaneously releasing the pressure in the side pouches on the bottom level and inflating the side pouches on the top level up to 90 kPa to restrict contraction and bending. This leads to a pure twisting mode generated from the bottom level, i.e., $[-1|\frac{0\leftrightarrow0}{1\leftrightarrow1}] \rightarrow \frac{BTC}{010}[-1|\frac{1\leftrightarrow1}{0\leftrightarrow0}]$. Because of the nonideal synchronization of releasing and inflating the side pouches, a small height change (5.14%) and discrepancy from the finite element simulations (Supplementary Fig. 12) are observed, which could be further improved in the future.

The coupled contraction and twisting mode is given in Fig. 3e and Supplementary Movie 6 with the following pressurization scheme, i.e., $\frac{BTC}{011}[-1|\frac{0\leftrightarrow0}{1\leftrightarrow1}]$, where one level of the module does not activate the side pouch (i.e., $0 \leftrightarrow 0$), leading to a coupled contraction and twisting mode (similar to Fig. 2b), and another level inflates the side pouches on both sides (i.e., $1 \leftrightarrow 1$) up to 90 kPa, leading to a very rigid unit to avoid cancellable twisting from the two levels with opposite chirality. Overall, a combined contraction and twisting mode is achieved. The simulation results (Supplementary Fig. 13) also agree reasonably well with the experiments.

Figure 3f and Supplementary Movie 7 show the results for the coupled bending and twisting, i.e., $\frac{BTC}{110}[-1|\frac{1\leftrightarrow0}{1\leftrightarrow1}]$, in which the asymmetric inflating of the side pouches in one level (i.e., $1 \leftrightarrow 0$) and vacuuming the main chamber resulting in coupled bending and twisting (similar to Fig. 2c), while keeping another level rigid by inflating all side pouches (i.e., $1 \leftrightarrow 1$) up to 90 kPa, which was also captured by the simulation (Supplementary Fig. 14).

The coupled contraction and bending mode is achieved using the following pressurization scheme $\frac{BTC}{101}[-1|\frac{1\leftrightarrow0}{1\leftrightarrow0}]$ (Fig. 3g, Supplementary Movie 8, and Supplementary Fig. 15). Although, at first glance, it is very similar to the pure bending mode (i.e., the same pressurization scheme), the pressure of 20 kPa in the side pouch is lower than that for pure bending (90 kPa); thus, the inflated side pouches do not provide enough constraints to prevent the contraction mode. The relationship between the degree of contraction and the pressure in the side pouch is shown in Supplementary Fig. 16 (see Supplementary Note 2) and is further explained in Supplementary Figs. 17 and 18 using finite element simulations. Note that the numbers (−1, 0, and 1) are used to denote the pressure states in pouches (atmospheric or positive pressure) and main chamber (atmospheric pressure or vacuum) instead of exact pressure values. Various deformation statuses inside the workspace can be exhibited by tuning the pressure in the main chamber and pouches. For example, the bending angle $\alpha = 17.8°$ when $P_p = 90$ kPa, $P_m = -4.5$ kPa (Supplementary Fig. 11); the bending angle $\alpha = 21.3°$ when $P_p = 20$ kPa, $P_m = -4.4$ kPa (Supplementary Fig. 15).

Figure 3h, Supplementary Movie 9, and Supplementary Fig. 19 show an all-three coupled mode, i.e., $\frac{BTC}{111}[-1|\frac{1\leftrightarrow0}{1\leftrightarrow0}]$, in which asymmetric inflating scheme $1 \leftrightarrow 0$ on one level leads to a coupled bending and twisting mode (see Fig. 2c), and inactive side pouches $0 \leftrightarrow 0$ result in a coupled twisting and contraction mode (see Fig. 2b). Consequently, this pressurization scheme $[-1|\frac{1\leftrightarrow0}{0\leftrightarrow0}]$ or $[-1|\frac{0\leftrightarrow0}{1\leftrightarrow0}]$ leads to an all-three coupled deformation mode.

A few remarks should be made for these seven distinct deformation modes. First, although the above cases involve only deflation in the main chamber, inflation can also be employed if the initial state is vacuumed, leading to an extension mode instead of the contraction mode. Second, the "|" symbol means that the pneumatic processes for the side pouches occur first, followed by those in the main chamber, which is also reflected in Fig. 3, where the operations on the side pouches lead the pneumatic processes. Third, although the pressurization scheme seems complicated, a certain pattern is rather clear. Upon relatively large pressure in the side pouches (e.g., 90 kPa), the symmetric scheme between the top and bottom levels leads to twisting-free modes because of the cancellable twisting mode, and vice versa; the symmetric scheme for the side pouches on either level leads to bending-free modes because of the symmetry, and vice versa. The output force of an origami module is tested (Supplementary Note 3) and shown in Supplementary Fig. 20.

## Complicated motions by connecting multiple origami modules in series

In addition to seven coupled and decoupled deformation modes of the present origami modules, its modular feature leads to a reconfigurable structure that is readily achievable through serially connecting multiple modules, endowing more complicated motions and better manipulation capabilities. Due to the inclusion of more local motion modes (i.e., twisting, contraction/extension), the present origami modules enable a highly dexterous structure, enlarging the workspace (see Supplementary Fig. 21), reducing the required modules, and simplifying the control of conventional continuum robots mainly consisting of multiple revolute joints or bending modules[24,44,45]. Figure 4a shows a robotic arm consisting of three origami modules connected in series, with a vacuum-driven gripper (see Methods, Supplementary Note 4 and Supplementary Fig. 22 for details) fixed on its distal end. The bolt-nut structure on the end cap connects each of the two modules and forms a pneumatic channel for both main chambers with negative pressure and side pouches with positive pressure. And the corresponding control system is shown in Supplementary Fig. 23 (see Methods). Figure 4b demonstrates the control sequence and the real-time images (see Supplementary Movie 10 for the whole process) of a robotic arm picking up a cup of water and pouring the water onto the target. Modules 1 and 2 first bend in the XZ plane toward the cup (steps 1 and 2); meanwhile, module 3 first bends up (i.e., bends in the XY plane) to avoid collision with the cup in step 1 and then bends down to grasp it in step 2. After grasping the cup (step 3), the three modules extend to return to the initial position in step 4, followed by step 5, in which module 2 twists to rotate the cup, along with a coupled twisting/contraction of module 1 to reduce the distance between the cup and the target. Meanwhile, module 3 bends in the XY plane to pour the water out of the cup. Similar to the motion modes in Fig. 4b, the three-module robotic arm can also water the grass along the length direction with coupled motion, i.e., twisting-contraction motion (see Supplementary Fig. 24a and Supplementary Movie 11).

An as-fabricated robotic arm is usually affected by its fixed workspace. Although origami designs can generally obtain a high contraction ratio to enlarge their workspace, limitations still exist when an object is beyond its working boundaries (Fig. 4c, Supplementary Movie 12). The plug-and-play characteristics of the present origami aim to solve this challenge. An additional origami module can be readily assembled onto the proximal end of the original robotic arm to construct a four-module structure (Fig. 4d, and Supplementary Movie 13), forming an amplified workspace (see Supplementary Note 5 and Supplementary Fig. 21) capable of reaching the object without redesigning and refabricating a totally new structure (Fig. 4e). The control sequence and the real-time images of the four-module origami arm manipulating a cup of water are shown in Fig. 4f (see Supplementary Movie 14 for the whole process). It is noteworthy that the demonstrated motions are customized for the specific application of pouring water from the cup and that more kinds of motions can be achieved via various combinations of deformation modes of each module (Fig. 5a and Supplementary Movie 15).

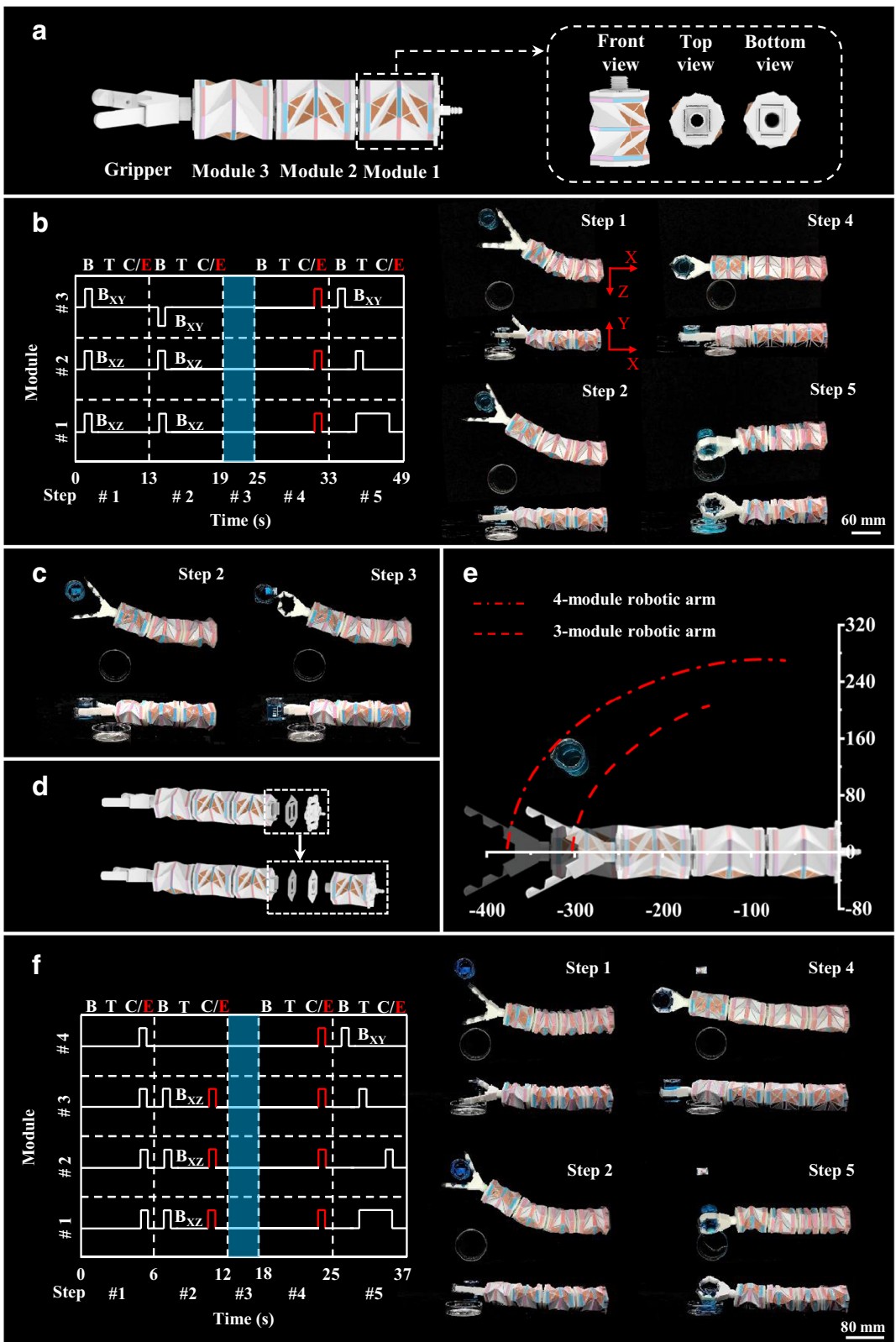

**Fig. 4 | Robotic arm with multiple modules. a** Schematic design of a three-module robotic arm and a module. The three-module robotic arm consists of three modules and a gripper (vacuum-actuated). **b** Diagram of the three-module robotic arm's motion modes and experimental results of the three-module robotic arm cap-grasping and pouring water from the top view and front view. **c** Diagram of the robotic arm failing to reach the cap when the cap is moved off the working boundary of the three-module robotic arm. **d** Process of a three-module robotic arm changing to a four-module robotic arm. **e** The partial working boundary of a three-module/four-module robotic arm. **f** Diagram of the four-module robotic arm's motion modes and experimental results of the four-module robotic arm cap-grasping and pouring water from the top view and front view. The shaded region in **b** and **f** indicates the grasping process of the soft gripper. C and E in C/E represent contraction and elongation, respectively.

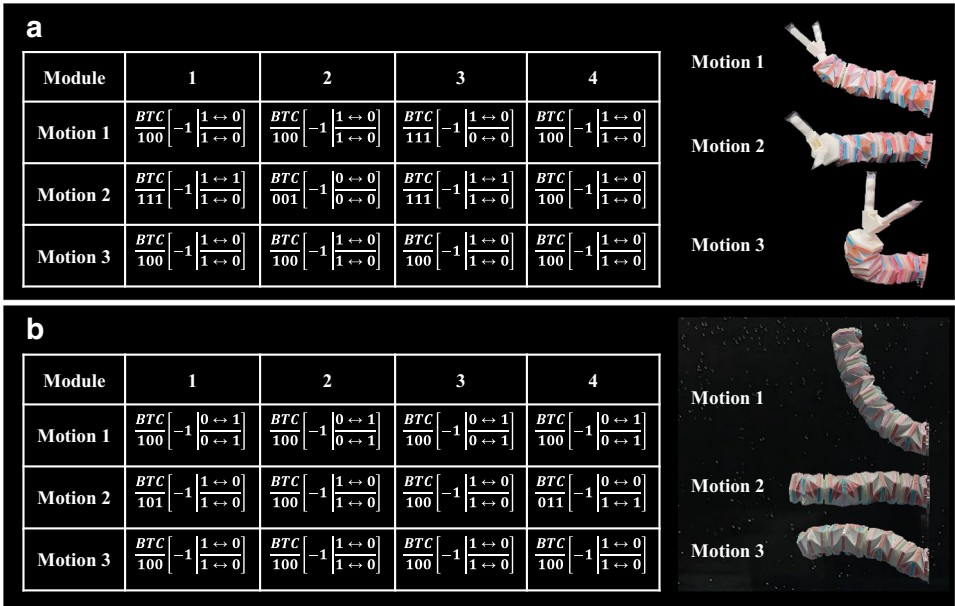

**Fig. 5 | Various motions of the robotic arm under different pressurization schemes. a** Pneumatic-driven spatial motion. **b** Liquid-driven underwater motions.

## Discussion

The all-purpose origami-inspired actuation module introduced here breaks the energy-favorable path of the conventional Kresling origami pattern, achieving seven distinct motion modes (i.e., bending, twisting, contraction/extension, and four combinations of these three basic types) through a single module. The decoupled, freely switchable motion modes make the present origami module outperform existing multimodal actuators with coupled motion types restricted by their mechanical structures or actuation distributions; moreover, its plug-and-play characteristics result in a reconfigurable structure that is readily connected in series and reassembled, further enlarging its workspace and functionality. The design, fabrication, working principle, and characteristics have been comprehensively discussed, making this origami module readily replicable in conjunction with its control strategies. Further improvements can be achieved by employing lightweight materials with higher strength and integrating sensing elements to build a more powerful robot with real-time self-perception. The fluidic-driven, multimodal motion also carries a high potential for specific applications, such as underwater manipulation (i.e., replacing the pneumatic source with water, shown in Fig. 5b and Supplementary Movie 16).

## Methods

### Geometric design

The key parameter to designing the Kresling origami pattern is the planar angular fraction λ (Supplementary Fig. 1), which affects the motion range and the stability of the Kresling origami module[34]. In this work, we chose λ = 0.58 to make the prototype in which the Kresling structure exhibits monostable, rendering the deformation of the Kresling structure continuously controllable by vacuum and readily recovering to its initial state after vacuuming. The compression test of a prototyped Kresling origami module (without pneumatic actuation) is investigated on the universal testing machine (JHY-5KN, Yuemang Intelligent Equipment Co., Ltd., China). The experimental and simulated results are shown in Supplementary Fig. 2. The experimental result shows good agreement with the simulated one and reveals that the designed Kresling structure is monostable.

### Fabrication of the origami module

To fabricate the origami module with two levels, we utilized Scotch tape with a thickness of 0.1 mm for the creases and polyvinyl chloride (PVC) panels with a thickness of 0.3 mm for the triangular facets in the Kresling pattern. The stiffness gradient between the PVC panels and Scotch tape creases enables the deformation of the structure, mainly at creases in twisting and contraction modes, while allowing the PVC panels to deform in bending modes. The Kresling patterns (one level or two levels) were cut on adhesive cutting mats using Silhouette CAMEO 4 (Supplementary Figs. 3a and 4a). Then, the colored paper was attached to the front side of the PVC panels, and another piece of Scotch tape was used to cover the colored paper (Supplementary Figs. 3b and 4b). The colored paper enables the deformation modes to be more visible. Next, the Scotch tape, PVC panels, and colored paper were peeled off from the adhesive cutting mat (Supplementary Figs. 3c and 4c). The inflatable side pouches were made of thermoplastic polyurethane (TPU) film with a thickness of 0.3 mm and were sealed with a heat-sealing machine. An air tube with a diameter of 1 mm was inserted into the pouch for inflation and deflation. The pouches were attached on the back side of the PVC panels by another piece of Scotch tape (Supplementary Figs. 3d and 4d). A two-level Kresling pattern was formed by rolling the sandwiching structures (Supplementary Figs. 3e and 4e). Finally, V-shaped grooves, an Ecoflex layer for air sealing, and T-shape clips were placed on both ends of the two-level pattern for ready assembly with another module (Supplementary Figs. 3f and 4f).

### The theoretical model to characterize the relationship between the pneumatic pressure and the origami pattern

The geometry of a Kresling pattern has been extensively studied[46]. Here, the focus was to derive the relationship between the deformed shape of a Kresling pattern (Supplementary Fig. 6a) and applied pneumatic pressure. The following assumptions were made: (1) the deformation energy of a Kresling pattern is localized at the creases (made of Scotch tape) and can be modeled as torsional springs, and (2) the work done by the pressure is stored in the Kresling pattern as torsional energy at the creases. The energy method was used to characterize the equivalent stiffness $k$ of the torsional spring. The bending moment $M$ imposed at the creases is given by $M = \frac{2E_{tape}t^3\psi}{3l}$[47], where $E_{tape}$ is the modulus of the Scotch tape, $2t$ (= 0.1 mm) is the thickness of the crease, $l$ (= 2 mm) is the width of the crease, and $\psi$ is the bending angle of the crease during deformation, which can also be obtained by $M = k\psi$ (see Supplementary Fig. 6b). Thus, the stiffness of

torsional spring $k$ is obtained as $k = \frac{2E_{tape}t^3}{3l}$. Supplementary Fig. 6c shows a one-level Kresling pattern with three types of creases, namely, 1, 2, and 3, with the corresponding initial equilibrium angles $\psi_1^0$, $\psi_2^0$, and $\psi_3^0$, respectively. Thus, the torsional energy stored in these creases is given by $W_1 = \frac{1}{2}|BD|k(\psi_1 - \psi_1^0)^2$, $W_2 = \frac{1}{2}|BC|k(\psi_2 - \psi_2^0)^2$, and $W_3 = \frac{1}{2}|CD|k(\psi_3 - \psi_3^0)^2$. Upon pneumatic pressure $P$, the work done by the pressure $P$ is given by $P\Delta V$, where $\Delta V$ is the volumetric change in the Kresling pattern, which can be determined by the geometrical relations. The energy stored at the creases is $6(W_1 + W_2 + 2W_3)$. By equating these two and plugging in the geometrical parameters (e.g., $a = b = 27.6$ mm), the relationship between the deformed shape of the Kresling pattern and the applied pressure was obtained and is shown in Fig. 2b.

### Details for the finite element simulations

We utilized the nonlinear finite element method to investigate the relationship between the performance[48] (i.e., height, torsional angle, and bending angle) and the actuation pressure (Supplementary Figs. 7–15 and Supplementary Figs. 17–19). The finite element model was established in ABAQUS (Dassault System). Linear elasticity models with different elastic moduli ($E$) and Poisson's ratios ($v$) were selected to describe the mechanical properties of the PVC ($E$: 1700 MPa; $v$: 0.3), TPU ($E$: 38 MPa; $v$: 0.4), and Scotch tape ($E$: 800 MPa; $v$: 0.4). The shell element S4R was used in the simulation for these materials. The plates on both ends of the origami module were established as rigid bodies in the finite element model. In addition, general contact was also considered in the finite element model. We employed an explicit dynamic method to analyze the deformation process of the origami modules. The computing time and accuracy of the whole model are mainly dependent on the size of the mesh. Thus, many simulations were performed to adjust the size of the mesh, and the stability of the simulation results was evaluated. After adjusting the mesh and evaluating the results, the minimum size of the mesh was set to approximately 0.33 mm.

### Fabrication of the vacuum-actuated gripper

The soft vacuum-actuated gripper is fabricated according to the following steps (Supplementary Fig. 22):

Step 1: thermoplastic polyurethane (TPU, ESUN, Xiaogan Esun New Material Co., Ltd., China) is used to 3D print the finger with the printer Raise3D Pro3 (Shanghai Fusion Tech Co., Ltd., China).

Step 2: a piece of TPU film is used to wrap the finger;

Step 3: the edges of the TPU film are sealed with a heat-sealing machine (Deli 16499, Deli Group Co., Ltd., China), followed by inserting a pneumatic tube into the TPU film.

Step 4: two identical vacuum-actuated fingers are inserted into the connector to form a soft gripper, which has a threaded hole and can connect with the origami module.

### Pneumatic control

The pneumatic control system consists of two parts: the pneumatic supply and control systems (Supplementary Fig. 23). The positive pressure was provided by a pressure pump and regulated by a precision regulator (IRV20-C06, SMC Inc.) powered by a regulated 24-volt supply. The vacuum was provided by a vacuum pump and regulated by a precision regulator (IR1000-01-A, SMC Inc.) powered by a regulated 24-volt supply. Solenoid valves V114-5LZB (SMC Inc.) and VK332V-5G-01 (SMC Inc.) are used to control the positive pressure in pouches and the vacuum in the main chamber, respectively. All the solenoid valves are located between precision regulators and the robotic arm, and connected with them through pneumatic tubes. The switches of the solenoid valves are controlled by the signal from the microcontroller (Arduino mega 2560, Arduino Inc.) via relays (TOUGLESY, Batu Na Group Co., Limited) powered by a 24-volt supply.

## Data availability

The data that support the findings of this study are available in Supplementary Information. Source data are provided in this paper.

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

## Acknowledgements

We thank the Research Center for Industries of the Future (RCIF) at Westlake University and Westlake Education Foundation for supporting this work. Z.Z. acknowledges support from the National Natural Science Foundations of China (Grant 52205031). Y.X. acknowledges support from the National Natural Science Foundation of China (Grants 91748209 and 11402229).

## Author contributions

C.Z., Z.Z., Y.P., and H.J. designed the experiments. C.Z., Z.Z., Y.P., Y.Z., S.A., Y.W., Zi.Z., Y.X., and H.J. carried out experiments and analyses. C.Z., Z.Z., and H.J. wrote the paper.

## Competing interests

The authors declare no competing interests.
