## [Peer Review File · Nature Communications]

REVIEWER COMMENTS

Reviewer #1 (Remarks to the Author):

In this study, the authors created a pneumatics-driven, origami-based soft robotic module that can show seven different deformations (aka. contraction, bending, twisting, and any combination of these three). By assembling these versatile modules, the authors fabricated soft robotic arms to perform complex manipulations.

Kresling origami has been used for soft robotic applications in quite a few studies before. Still, this paper successfully shows seven deformation modes for the first time using extensive experimentation, finite element simulation, and analytical modeling. This is quite an achievement and worthy of publication at Nature Communications. However, I do have a few recommendations for further improvement/clarification.

The first comment concerns Kresling's design: The geometry of the triangular facets on Kresling's side provides the kinematics underpinning of its contraction-twisting-bending coupling. Therefore, it is usually treated as a critical design parameter. However, the authors presented their Kresling design as is. It would be nice to elaborate more on the rationale behind their design selections. What would be the optimal Kresling design (if any) with the most balanced bending, twisting, and contraction? Or are there any Kresling designs that are easier to decouple twisting from bending and contraction? Based on the supplement videos, it seems that the authors chose the Kresling design solely because the hexagon end plates would align again once it fully contracts.

The second comment concerns the actuation method. In this study, the pneumatic actuation is simply on or off (-1, 0, or 1). I felt that this significantly constrained the potential of the dual-Kresling module. Can we expand the achievable kinematic space of the Kresling arm by fine-tuning the pressure control? That is, rather than supplying discrete pressure (at -1, 0, or 1), we can control the pressure to be anywhere between -1 and 1. Based on the authors' experiment and finite element results, the correlations between different deformation modes and input pressure are pretty nonlinear. And it is unclear if the twisting-contraction decoupling would work if the pressure is low.

The third comment is about the assembly of Kresling modules. By carefully looking at the different videos of robotic manipulation, these modules seem to be attached at different orientations (with respect to its axial direction) to fine-tune the bending directions. This is probably because every model bends only in one direction along the pressurized side pouch. Therefore, can the authors shed more light on how the axial orientation of the Kresling module is determined based on the targeted manipulation tasks?

Reviewer #2 (Remarks to the Author):

In this paper, the authors propose a plug and play module that allows for diverse actuation modes in an origami structure. By adding two creases and an additional pouch, the Kresling origami structure can achieve three decoupled actuation modes: bending, twisting and contraction/expansion. Furthermore, by combining these basic modes, the actuator can achieve another four additional modes. The authors fabricate and test these origami-based structures, and then demonstrate their applications in air and water. The experimental results show that the proposed design can greatly expand the versatility of origami structures.

This idea for multi-functional origami structures is very interesting. The proposed design can also significantly expand the workspace of an origami-based robot, leading to wider practical applications. The authors may want to consider the following comments to revise the manuscript.

1. Since various actuators (including electric motors and other smart actuators) can achieve actuation modes of bending, twisting and contraction, the authors may want to discuss the unique attributes of this origami structure with more details.
2. Does viscoelasticity play a role? Is the performance of the origami structure affected by the deflation speed of air?
3. For any actuators, the output force is significant in practical applications. More discussion on the output force of this origami structure can be important and interesting.
4. While the authors show the effects of air pressure in the main chamber on the deformation of the origami unit, how about the effect of air pressure in the pouches? Can the origami unit's behavior be tuned by adjusting the pressure in the pouches?
5. It is mentioned that the origami arm's various deformation modes can expand the workspace. The evaluation of its workspace can be interesting.
6. In Figure 3, some words are too small to identify. In Figure 4, the pictures of the origami arm can be larger for clear viewing.
7. It is interesting to see the operating of the origami arm in air and water. How the different environmental conditions affect the performance of the origami structures?

Reviewer #3 (Remarks to the Author):

This paper presented pneumatic-driven origami modules with different deformation modes. The design is based on the Kresling origami pattern. The physical implementations of the design and demonstrations were well-developed and presented in the supplemental videos. The plug-and-play origami modules could be combined to generate complex motions. The presented technical work is interesting and practical and shows the potential for broad applications. However, several major concerns are related to the originality, inaccurate discussion provided for existing works, and written presentation in the current manuscript. Please find below for detailed comments.

Comments:

The major concern I have is the novelty of the presented work. The adopted origami design has been previously proposed and demonstrated for its diverse motion capabilities. The design methodology was presented in [Sung & Rus 2025], and the design implementation was detailed in [Wu et.al. 2021]. Please see below for the references:

Sung, C., & Rus, D. (2015). Foldable joints for foldable robots. *Journal of Mechanisms and Robotics*, 7(2), 021012.

Wu, S., Ze, Q., Dai, J., Udipi, N., Paulino, G. H., & Zhao, R. (2021). Stretchable origami robotic arm with omnidirectional bending and twisting. *Proceedings of the National Academy of Sciences*, 118(36), e2110023118.

The first sentence of the abstract does not clarify deformation modes of what – it must specify what this paper is about and the deformation it refers to [Line 23].

Actuation is an important aspect of the presented work, but I cannot find any information about pneumatic actuation. All videos show the actuation part hidden. I would like to see how the modules are individually inflated and controlled.

The literature review does not seem to reflect the previous work correctly. For example, the work presented in [12] shows the twisted tower mechanism, which can contract, bend, and twist, while the authors stated it as a unimodal motion [Line 41]. I suggest the authors carefully revisit this and other references for existing mechanisms and their feasible motions.

The authors claim that the modules are soft, but most structural components appear rigid, with deformable creases.

For the first reviewer:

In this study, the authors created a pneumatics-driven, origami-based soft robotic module that can show seven different deformations (aka. contraction, bending, twisting, and any combination of these three). By assembling these versatile modules, the authors fabricated soft robotic arms to perform complex manipulations. Kresling origami has been used for soft robotic applications in quite a few studies before. Still, this paper successfully shows seven deformation modes for the first time using extensive experimentation, finite element simulation, and analytical modeling. This is quite an achievement and worthy of publication at Nature Communications. However, I do have a few recommendations for further improvement/clarification.

We thank the reviewer for the positive feedback and the following valuable comments, which helps us to further improve the quality of this manuscript.

1. The first comment concerns Kresling's design: The geometry of the triangular facets on Kresling's side provides the kinematics underpinning of its contraction-twisting-bending coupling. Therefore, it is usually treated as a critical design parameter. However, the authors presented their Kresling design as is. It would be nice to elaborate more on the rationale behind their design selections. What would be the optimal Kresling design (if any) with the most balanced bending, twisting, and contraction? Or are there any Kresling designs that are easier to decouple twisting from bending and contraction? Based on the supplement videos, it seems that the authors chose the Kresling design solely because the hexagon end plates would align again once it fully contracts.

Thanks for the comment. The key parameter determining the motion range and the stability of the Kresling origami module is the ratio λ between the angles $\angle FCD$ and $\angle OAB$ (**Fig. S1**). In this work, we chose $\lambda = 0.58$ to make the prototype. For this structure, Kresling structure is monostable. Further discussions about the geometric design of the Kresling origami structure are now included in **Supplementary Note 1**. In addition, to decouple twisting from bending and contraction, one can actively control the pneumatic pressure. The decoupled motion has no direct correlation with the geometric parameters.

The newly added text, figures, and corresponding captions are:

The geometric design of the one-level origami structure can be seen in Supplementary Note 1, Supplementary Fig. 1, and Supplementary Fig. 2. The detailed fabrication processes and the pneumatic actuation methods are provided in Supplementary Fig. 3, Supplementary Fig. 4, and Supplementary Fig. 5.

Supplementary Note 1. Geometric design

The key parameter to design the Kresling origami pattern is the planar angular fraction λ (Supplementary Fig. 1), which affects the motion range and the stability of the Kresling origami module³⁴. In this work, we chose $\lambda = 0.58$ to make the prototype, in which the Kresling structure exhibits monostable, rendering the deformation of the Kresling structure continuously controllable by vacuum and readily recovering to its initial state after vacuumizing. The compression test of a prototyped Kresling origami module (without pneumatic actuation) is investigated on the universal testing machine (JHY-5KN, Yuemang Intelligent Equipment Co., LTD, China). The experimental and simulated results are shown in Supplementary Fig. 2. The experimental result shows good agreement with the simulated one and reveals that the designed Kresling structure is monostable.

Supplementary Fig. 1. Geometry of the one-level Kresling structure. λ is the ratio between the angle $\angle FCD$ and $\angle OAB$.

Supplementary Fig. 2. Experimental and simulated results of the compression test.

2. The second comment concerns the actuation method. In this study, the pneumatic actuation is simply on or off (-1, 0, or 1). I felt that this significantly constrained the potential of the dual-Kresling module. Can we expand the achievable kinematic space of the Kresling arm by fine-tuning the pressure control? That is, rather than supplying discrete pressure (at -1, 0, or 1), we can control the pressure to be anywhere between -1 and 1. Based on the authors' experiment and finite element results, the correlations between different deformation modes and input pressure are pretty nonlinear. And it is unclear if the twisting-contraction decoupling would work if the pressure is low.

Thanks for this in-depth comment.

Regarding the first part of this comment, though we used the numbers (-1, 0, and 1) to denote the pressure states in pouches (atmospheric or positive pressure) and the main chamber (atmospheric pressure or vacuum), we did study the effect of pressure on motion modes. For example, the bending angle is $\alpha = 17.8^\circ$ when $P_p = 90$ kPa, and $P_m = -4.5$ kPa (Supplementary Fig. 11); and becomes to $\alpha = 21.3^\circ$ when $P_p = 20$ kPa and $P_m = -4.4$ kPa (Supplementary Fig. 15).

The added text reads:

Note that the numbers (-1, 0, and 1) are used to denote the pressure states in pouches (atmospheric or positive pressure) and main chamber (atmospheric pressure or vacuum) instead of exact pressure values. Various deformations inside the workspace can be exhibited via tuning the pressure in the main chamber and pouches.

For example, the bending angle is $\alpha = 17.8^\circ$ when $P_m = 90$ kPa and $P_m = -4.5$ kPa (Supplementary Fig. 11); and becomes to $\alpha = 21.3^\circ$ when $P_m = 20$ kPa and $P_m = -4.4$ kPa (Supplementary Fig. 15).

Regarding the second part of this comment, we have conducted new experiments to address this point. As shown in Fig. R1, when the pressure value in pouches is low (20 kPa and 40 kPa), the twisting-contraction decoupling could also be achieved. The pressure decrease will not affect the realization of decoupling, but will lead to a smaller motion range if the pressure is too low (less than 10 kPa).

Fig. R1. Decoupled twisting motion of the origami module when (a) $P_p = 20$ kPa and (b) $P_p = 40$ kPa.

3. The third comment is about the assembly of Kresling modules. By carefully looking at the different videos of robotic manipulation, these modules seem to be attached at different orientations (with respect to its axial direction) to fine-tune the bending directions. This is probably because every model bends only in one direction along the pressurized side pouch. Therefore, can the authors shed more light on how the axial orientation of the Kresling module is determined based on the targeted manipulation tasks?

Appreciate this comment. The axial orientation of each module depends on the prescribed motion trajectory. For example, the pouch arrangement of all modules should be the same for planar motion, while one or two modules can be rotated 90° for out-of-plane movements. At present, two side pouches are integrated inside each origami module for the sake of ease of fabrication. In order to achieve omnidirectional movement automatically, all six sides of the present origami module should be integrated with pouches. Thus, no more tuning of axial orientation is needed.

For the second reviewer:

In this paper, the authors propose a plug and play module that allows for diverse actuation modes in an origami structure. By adding two creases and an additional pouch, the Kresling origami structure can achieve three decoupled actuation modes: bending, twisting and contraction/expansion. Furthermore, by combining these basic modes, the actuator can achieve another four additional modes. The authors fabricate and test these origami-based structures, and then demonstrate their applications in air and water. The experimental results show that the proposed design can greatly expand the versatility of origami structures.

This idea for multi-functional origami structures is very interesting. The proposed design can also significantly expand the workspace of an origami-based robot, leading to wider practical applications. The authors may want to consider the following comments to revise the manuscript.

We are grateful to the reviewer for the positive feedback.

1. Since various actuators (including electric motors and other smart actuators) can achieve actuation modes of bending, twisting and contraction, the authors may want to discuss the unique attributes of this origami structure with more details.

Thanks for the suggestion. Various motion modes like contraction, bending, and twisting can indeed be achieved by various actuators; however, most actuators barely exhibit a single motion mode with a single module. Even though some novel designs can achieve multiple modes through a single module, the available modes are still limited and generally coupled. Therefore, the advantage of our design is mainly the ability of all seven deformation modes through one module. We added new text in the revised Introduction and the text reads:

Most actuators, including conventional electric motors or smart materials (**Fig.1b**), generally exhibit single motion mode. Even though some designs can achieve multiple modes through a single module, the available modes are still limited and generally coupled. In contrast, our design achieves the integration of all seven motion modes through barely one module.

2. Does viscoelasticity play a role? Is the performance of the origami structure affected by the deflation speed of air?

Thanks for this excellent comment. As we used tape to make the creases, there is viscoelasticity effect.

The viscoelasticity of the creases and the air in the main chamber are related to the loading rate. In this work, the loading method is the vacuum actuation. When the loading rate is relatively small, the performance of a module mainly manifests as elasticity. However, when the loading rate is relatively high, the deformation of a module become complex, due to the viscoelasticity of the creases and the air as well as the dynamic response of the structure. However, the final states (i.e., deformation modes) do not depend on the rate of air flow.

We have conducted experiments to study the deformation of a module with different deflation speed in contraction and bending modes (Fig. R2). Though the dynamic process depends on the rate of air flow, the final states (i.e., deformation modes) do not depend on the rate of air flow. The air flow is tested with flowmeter (MF5708, Siargo Co., LTD. Germany). The pressure is tested with gas-pressure meter (DP-101, Panasonic Co., LTD. Japan).

Fig. R2. Deformation of a module with different deflation speed in contraction (a) and bending motion (b).

We added this figure as Supplementary Fig. 25. The caption is

Supplementary Fig. 25. Deformation of a module with different deflation speed in contraction (a) and bending motion (b). The air flow is tested with flowmeter (MF5708, Siargo Co., LTD. Germany). The pressure is tested with gas-pressure meter (DP-101, Panasonic Co., LTD. Japan). The results show that the dynamic process depends on the rate of air flow, but the final states (i.e., deformation modes) do not depend on the rate of air flow.

3. For any actuators, the output force is significant in practical applications. More discussion on the output force of this origami structure can be important and interesting.

Appreciate this excellent comment. According to the reviewer's suggestion, we have conducted several tests on the present origami module, measuring its contraction force, bending force, and twisting moment. A new figure is included in the revised supplementary information, with the relevant descriptions in Supplementary Note 4.

The newly added text, figure, and corresponding caption are:

The measurements and results of the output contraction force, bending force, and twisting moment are shown in Supplementary Fig. 20. The force sensor (Mini 45, ATI Industrial Automation Co., LTD, USA) is fixed on a fixed beam. The bottom of the origami module is fixed on the platform, and the top of the module is connected to the force sensor via a plastic bolt. During the contraction process (Supplementary Fig. 20a), the tension force F_c increases with the increase of the vacuum value, and the maximum force is 22.9 N (when $P_m = -8$ kPa). As to the bending process (Supplementary Fig. 20b), the shearing force F_b also becomes larger with the increase of the vacuum value in the main chamber; the maximum value of F_b is 1.91 N (when $P_m = -12$ kPa). In the first stage (pressurizing bottom pouches) of the twisting motion (Supplementary Fig. 20c), the torque is along the negative direction of the Z-axis. However, in the second and third stages, the direction of the torque changes to the positive direction of the Z-axis. The maximum value of T_i is 0.21 Nm.

Supplementary Fig. 20. Measurements of the output contraction force (a), bending force (b), and twisting moment (c). F_c and F_b are the force along the Z-axis and X-axis, respectively. T_i is the torque along the Z-axis.

4. While the authors show the effects of air pressure in the main chamber on the deformation of the origami unit, how about the effect of air pressure in the pouches? Can the origami unit's behavior be tuned by adjusting the pressure in the pouches?

Thanks for the comment. The pressure in pouches affects the behavior of the origami module, which has been included in the original submission. For example, the bending angle is $\alpha = 17.8^\circ$ when $P_m = 90$ kPa and P_m

= -4.5 kPa (Supplementary Fig. 11); and becomes to $\alpha = 21.3^\circ$ when $P_m = 20$ kPa and $P_m = -4.4$ kPa (Supplementary Fig. 15). We added new text to make it more clearly:

Various deformations inside the workspace can be exhibited via tuning the pressure in the main chamber and pouches. For example, the bending angle is $\alpha = 17.8^\circ$ when $P_m = 90$ kPa and $P_m = -4.5$ kPa (Supplementary Fig. 11); and becomes to $\alpha = 21.3^\circ$ when $P_m = 20$ kPa and $P_m = -4.4$ kPa (Supplementary Fig. 15).

5. It is mentioned that the origami arm's various deformation modes can expand the workspace. The evaluation of its workspace can be interesting.

The twisting motion does not affect the working space in plane XOZ; therefore, only the working boundary of bending and contraction motion are discussed in the following texts and shown in Supplementary Fig. 20, which will be added in the Supplementary Information.

We added some new text in Supplementary Note 8 to address the working boundary. The new text reads:

Conventional Kresling actuator can only contract and twist along its axial direction, making its workspace a single straight line, as shown in Supplementary Fig. 21c and Supplementary Fig. 21d (orange line). However, after selectively breaking the energy favorable deformation mode of the Kresling origami pattern, the actuator we obtained can exhibit bidirectional bending (the design with two side pouches), leading to an expansion of its workspace from a straight line to a planar area (irregular polygon ADE in Supplementary Fig. 21c). Besides, after the 3-module robotic arm is assembled with another module forming a 4-module robotic arm, its workspace can be further expanded (irregular polygon FLM in Supplementary Fig. 21d). The black solid lines, red dash-dot lines and blue dash-dot lines (in Supplementary Fig. 21c and Supplementary Fig. 21d) can be calculated as the following equations (Eq. S7 and Eq. S8). The green solid lines in Supplementary Fig. 21c and Supplementary Fig. 21d are fitted with the points A, B, C and D as well as F, G, J and L, respectively.

$$x = - \left(\sum_{i=0}^{m-1} [H' \cos(\gamma + i\alpha)] + |BC| \cos(m\alpha) \right) \quad (S7)$$

$$y = \sum_{i=0}^{m-1} [H' \sin(\gamma + i\alpha)] + |BC| \sin(m\alpha) \quad (S8)$$

where m represents the amount of bending module. The analysis of Fig. 4e does not consider the planar bending of the distal module, in accordance with the real working status (pouring water from the cup) of the prototyped robotic arm. Note that the analysis in this note is based on the design with two side pouches integrated inside each origami module for the sake of ease of fabrication. If all six sides of the present origami module are integrated with pouches, omnidirectional movement can be achieved, then forming a spatial workspace (the current planar one spanning around the axial direction).

Supplementary Fig. 21. Workspace of the 3-module and 4-module robotic arms. The relationship between the bending angles γ and α , as well as the relationship between γ and H' in the condition of $P_p = 20$ kPa (a) and $P_p = 65$ kPa (b). The workspace of the 3-module (c) and 4-module (d) robotic arms. The black solid lines, red dash-dot lines and blue dash-dot lines are the working boundaries of the 3/4 module arms bending in the condition of $P_p = 90$ kPa, $P_p = 65$ kPa and $P_p = 20$ kPa, respectively. The orange solid lines are the working boundaries when the arms contract.

6. In Figure 3, some words are too small to identify. In Figure 4, the pictures of the origami arm can be larger for clear viewing.

Thanks to the reviewer's reminder. Figs. 3 and 4 are revised as follows:

Fig. 3. Design concept and motion capability of an all-purpose origami module.

Fig. 4. Robotic arm with multiple modules.

7. It is interesting to see the operating of the origami arm in air and water. How the different environmental conditions affect the performance of the origami structures?

Thanks for the comment. The resistance of the origami arm is larger in the underwater condition, compared with the one on the ground. Thus, the response of the underwater origami arm is relatively slow. Nevertheless, the deformation ranges under the two different environmental conditions are the same due to the balance of internal and external pressures.

For the third reviewer:

This paper presented pneumatic-driven origami modules with different deformation modes. The design is based on the Kresling origami pattern. The physical implementations of the design and demonstrations were well-developed and presented in the supplemental videos. The plug-and-play origami modules could be combined to generate complex motions. The presented technical work is interesting and practical and shows the potential for broad applications. However, several major concerns are related to the originality, inaccurate discussion provided for existing works, and written presentation in the current manuscript. Please find below for detailed comments.

We are grateful to the reviewer for the encouragement and constructive criticism, which motivate us to improve the quality of this work.

1. *The major concern I have is the novelty of the presented work. The adopted origami design has been previously proposed and demonstrated for its diverse motion capabilities. The design methodology was presented in [Sung & Rus 2025], and the design implementation was detailed in [Wu et.al. 2021]. Please see below for the references:*

*Sung, C., & Rus, D. (2015). Foldable joints for foldable robots. *Journal of Mechanisms and Robotics*, 7(2), 021012.*

*Wu, S., Ze, Q., Dai, J., Udipi, N., Paulino, G. H., & Zhao, R. (2021). Stretchable origami robotic arm with omnidirectional bending and twisting. *Proceedings of the National Academy of Sciences*, 118(36), e2110023118.*

Thanks for the comment.

The two papers that the reviewer mentioned utilized Kresling pattern to build robotic structures with certain motion capabilities. However, their designs are based on the classical Kresling pattern, making their motion modes limited by the Kresling origami structure, namely twisting-coupled contraction or bending. On the contrary, our work breaks the energy favorable deformation mode of the classical Kresling pattern through individually accessible, pneumatically driven pouches on the side of the pattern, and consequently, a two-level Kresling origami pattern with the opposite chirality was created as an all-purpose module with seven distinct motion modes. Therefore, the core novelty of this work is to improve the current, widely-used Kresling pattern and realize seven deformation modes via only one module, which outperforms existing works. These two papers have been cited in the revised manuscript (refs 19 and 28).

We have revised the manuscript to emphasize the core novelty, and the new text reads:

Even though some designs can achieve multiple modes through a single module, the available modes are still limited and generally coupled (**Fig.1b**). In contrast, our design achieves the integration of all seven motion modes through barely one module.

2. *The first sentence of the abstract does not clarify deformation modes of what – it must specify what this paper is about and the deformation it refers to [Line 23].*

Thanks to the reviewer's kind reminder. The deformation of an object should consist of three basic deformation modes, i.e., contraction/elongation, bending, and twisting. The first sentence of the abstract is now revised and reads:

Three basic deformation modes of an object (bending, twisting, and contraction/extension) along with their various combinations and delicate controls lead to diverse locomotion.

3. Actuation is an important aspect of the presented work, but I cannot find any information about pneumatic actuation. All videos show the actuation part hidden. I would like to see how the modules are individually inflated and controlled.

The inner details of the present module are shown in Supplementary Fig. 5.

Supplementary Fig. 5. Inner details of the present origami module. Each pouch consists of a TPU bag, a plastic strip, an air tube connector and a thin air tube. The plastic strip is utilized to maintain the air path when the pouch is folded. The air tube connector is used to link the air tube and the TPU bag. All the soft air tubes are placed in the main chamber. We inflate/deflate pouches through air tubes and deflate the main chamber via the vent bolt. Each pouch is linked with two solenoid valves (one for positive pressure control (V114-5LZB, SMC Inc.) and the other for vacuum control (VK332V-5G-01, SMC Inc.)) through thin air tubes. The pressure inside pouches is controlled by precision regulators (IRV20-C06, SMC Inc.). The main chamber is actuated by vacuum with a precision regulator (IR1000-01-A, SMC Inc.).

4. The literature review does not seem to reflect the previous work correctly. For example, the work presented in [12] shows the twisted tower mechanism, which can contract, bend, and twist, while the authors stated it as a unimodal motion [Line 41]. I suggest the authors carefully revisit this and other references for existing mechanisms and their feasible motions.

Appreciate this comment. We have checked the references, and the revision is in the following:

Consequently, actuators with multimodal modes have emerged for coupled motion types (e.g., coupled twisting and contraction)¹⁹⁻²⁸. Among these structures of actuators, such as twisting tower^{13, 19}, elastic sheets with kirigami patterns²⁰, local constraints^{11, 21-23}, parallel spaced multiple chambers²⁴⁻²⁶ and origami-inspired designs^{21, 27-31}, such as Yoshimura^{32, 33}, and Kresling origami^{34, 35} and Miura³⁶ represent a distinctive structural design method offering unique merits in actuation, including a high deformation ratio³⁶⁻³⁸, multi-stability^{30, 39}, and manipulative stiffness^{40, 41}, although existing origami-based actuators still suffer limited multi-degree-of-freedom motions resulting from their original origami patterns.

5. The authors claim that the modules are soft, but most structural components appear rigid, with deformable creases.

Thanks to the reviewer's comments. We understand the misleadingness of using "soft" in the manuscript and have removed "soft"-ness to describe the present module.

REVIEWERS' COMMENTS

Reviewer #1 (Remarks to the Author):

As I stated in the original report, the authors managed to achieve all seven deformation modes *independently* from the Kresling modules and demonstrated compelling applications (aka complex robotic manipulations). This is a substantial achievement. Also, the authors addressed my comments quite well with additional experiments and elaborations. I, therefore, recommend publication at Nature Communications.

Reviewer #2 (Remarks to the Author):

The revision satisfies this reviewer.

Reviewer #3 (Remarks to the Author):

The authors have improved the clarity and addressed the comments raised by the reviewers. One remaining concern is that significant technical contents are provided in the supplemental documents, while the main manuscript must be sufficient and complete as it is. Some technically important information, such as the design of the actuation mechanism, still need to be included in the main manuscript.

For the first reviewer:

*As I stated in the original report, the authors managed to achieve all seven deformation modes *independently* from the Kresling modules and demonstrated compelling applications (aka complex robotic manipulations). This is a substantial achievement. Also, the authors addressed my comments quite well with additional experiments and elaborations. I, therefore, recommend publication at Nature Communications.*

We would like to thank the reviewer for the positive feedback and recommendation.

For the second reviewer:

The revision satisfies this reviewer.

We would like to thank the reviewer for taking the time to carefully review our paper and helping us improve the paper quality.

For the third reviewer:

The authors have improved the clarity and addressed the comments raised by the reviewers. One remaining concern is that significant technical contents are provided in the supplemental documents, while the main manuscript must be sufficient and complete as it is. Some technically important information, such as the design of the actuation mechanism, still need to be included in the main manuscript.

Thanks for the suggestion. We have added the Supplementary Note 1, Supplementary Note 5, and Supplementary Note 7 into the Methods section.